# Evaluating Fluid Distribution by Distributed Acoustic Sensing (DAS) with Perforation Erosion Effect

**DOI:** 10.3390/s25227037

**Published:** 2025-11-18

**Authors:** Daichi Oshikata, Ding Zhu, A. D. Hill

**Affiliations:** Petroleum Engineering Department, Texas A&M University, College Station, TX 77843, USA; oshikata@tamu.edu (D.O.); danhill@tamu.edu (A.D.H.)

**Keywords:** distributed acoustic sensing, fiber-optic sensing, hydraulic fracturing, petroleum engineering, CFD simulation, perforation geometry

## Abstract

Among the various completion strategies used in multi-stage hydraulic fracturing of horizontal wells, the limited entry design has become one of the most common approaches to promote more uniform slurry distribution. This method involves reducing the number of perforations so that higher perforation friction is generated at each entry point. The increased pressure drops force fluid and proppant to be diverted across multiple clusters rather than concentrating at only a few, thereby enhancing stimulation efficiency along the lateral. In this study, Computational Fluid Dynamics (CFD) simulations were performed to investigate how perforation erosion influences acoustic signals measured by Distributed Acoustic Sensing (DAS). Unlike previous studies that assumed perfectly circular perforations, this work uses oval-shaped geometries to better reflect the irregular erosion observed in the field, which provides more realistic modeling. The workflow involved building wellbore and perforation geometries, generating computational meshes, and solving transient turbulent flow using Large Eddy Simulation (LES) coupled with the Ffowcs Williams–Hawkings (FW-H) acoustic model. Acoustic pressure was then estimated at receiver points and converted into sound pressure level for analysis. The results show that, for a given perforation size, changes in flow rate cause log(q) versus sound pressure level to follow a straight line defined by a constant slope and varying intercept. Even when erosion alters the perforation into an oval shape, the intercept increases logarithmically, resulting in reduced sound amplitude, while the slope remains unchanged. Furthermore, when the cross-sectional area and flow rate are equal, oval perforations produce higher sound amplitudes than circular ones, suggesting that perforation geometry has a measurable influence on the DAS signal. This indicates that even when the same amplitude DAS signal is obtained, assuming circular perforations when estimating the fluid distribution leads to an overestimation if the actual perforation shape is oval. These findings highlight the importance of considering irregular erosion patterns when linking DAS responses to fluid distribution during hydraulic fracturing.

## 1. Introduction

Although renewable energy and battery storage are rapidly gaining prominence in the U.S. power sector, petroleum—including unconventional resources—still accounts for a substantial share of total energy consumption. Consequently, the development of unconventional reservoirs remains essential, and continued optimization of their exploitation is required. However, unconventional formations typically suffer from challenges such as low permeability and thin pay zones, which necessitate engineering solutions to achieve economic production. A common approach is to drill long horizontal wells and apply multi-stage hydraulic fracturing [1]. In most developments, a plug-and-perf completion is employed: a casing is run and cemented in the annulus between the casing and the wellbore; perforations are then created, and slurry is injected to initiate fractures. After the first stage is completed, a bridge plug is set to isolate the interval, and the process is repeated for the subsequent stage. This sequence is typically performed for 30 or more stages along the lateral, hence the term plug-and-perf completion.

In this approach, achieving uniform fracture development across stages is critical to improving production. To promote a more even slurry distribution along the wellbore, recent practice has increased injection rates and the volumes of both proppant and fluid. This strategy aligns with the limited-entry completion concept, in which the number of perforations is intentionally reduced to increase the frictional pressure drop at each perforation and thereby ensure that all clusters receive sufficient fluid and proppant. However, because this design employs high injection rates and large proppant volumes, proppant impacts the casing walls, gradually enlarging the diameter, and this phenomenon is known as perforation erosion.

During hydraulic fracturing, initially eroded perforations experience reduced frictional resistance, leading to greater slurry intake and further erosion. As a result, by the end of treatment, stimulation is concentrated in only eroded perforations, reducing overall cluster efficiency. Field observations using downhole video cameras and ultrasonic imaging have shown that severe erosion occurs on the lower side of the wellbore, while minimal erosion is observed on the upper side [2,3,4]. This is mainly attributed to higher proppant concentration on the lower side due to gravity [5,6,7] and slight differences in initial perforation size caused by perforation gun clearance [3,8]. Regarding proppant distribution between the toe and heel clusters, the findings are inconsistent among studies: laboratory and numerical results show higher proppant concentration on the toe side [5,6,7], whereas field observations indicate more severe erosion on the heel side clusters [2,4]. 

Fiber optic sensing technologies provide a valuable tool to monitor slurry distribution, with distributed acoustic sensing (DAS) being one of the primary methods. During fracturing, DAS continuously records acoustic signals generated by fluid flow through the perforations, enabling observation of dynamic changes in fluid distribution. In the field case conducted in Oman where DAS was applied, an excellent match was observed between the fluid and proppant distributions obtained from the simulator and those derived from DAS/Distributed Temperature Sensing (DTS) monitoring performed during the hydraulic fracturing treatment [9]. This confirmed that DAS is an effective tool for fracture diagnostics. Furthermore, in another field application using high-frequency DAS data acquired during multi-cluster hydraulic fracturing operations, the behavior of individual clusters was successfully distinguished, stage-by-stage injection volumes were estimated, and the temporal evolution of fracture geometry was tracked [10]. Figure 1 [11] shows the DAS waterfall plot during hydraulic fracturing; the x-axis shows time, and the y-axis shows depth. The colors represent the acoustic signal intensity, and the warmer colors observed at the cluster depth (green triangle) indicate the fluid injection. In addition, the red dashed line and markers ①, ②, and ③ indicate the times when perforation images were taken using the downhole camera. Figure 1 shows that, as the treatment progresses, the DAS signal weakens and eventually disappears. 

The perforation images corresponding to markers ①, ②, and ③ are shown in Figure 2, where erosion and enlargement of the perforations can be observed. These observations indicate that, as erosion advances, the acoustic signal reduces. In addition, as Figure 2 indicates, perforations do not always erode into circular shapes but often take on irregular geometries. Although previous studies have investigated the influence of perforation erosion on DAS signals, those studies assumed circular perforation geometries, and the effect of irregular shapes on DAS signal response is poorly understood. Therefore, in this study, we reproduce the perforation geometry observed in the field using oval-shaped perforations and investigate its effect for more realistic DAS interpretation.

## 2. Acoustic Theories

### 2.1. Empirical Correlation of Acoustics Induced by Perforation Flow

Chen et al. [13] explored the impact of proppant concentration and particle size in a laboratory wellbore model. In the experiment, fracture cells filled with proppant were used and connected to the wellbore with pipe simulating perforation. Fluid was injected from the fracture cell, and the sound generated at the perforation was measured. Pakhotina et al. [14,15] validated similar experiments through numerical simulations with Computational Fluid Dynamics (CFD). Their combined findings established a linear correlation between the logarithm of flow rate cubed and the corresponding sound pressure level, showing good consistency between physical experiments and numerical modeling.(1)$$\log\left(q^{3}\right)=A\times L_{sp}+B$$
where q is the flow rate at a perforation, $L_{sp}$ is the measured sound pressure level, and A and B are the coefficients of the linear correlation. Using field DAS data from a fracturing stage with a single cluster and without proppant injection, Sakaida et al. [16] demonstrated that the acoustic correlation remains applicable when the sound pressure level is replaced by the frequency band energy, representing the energy response of the DAS signal. Sakaida et al. [16] applied this acoustic correlation to estimate cluster-level flow rates from DAS measurements.

### 2.2. Perforation Erosion Effect on Acoustic Signal

Hamanaka [17] conducted numerical simulations using CFD to investigate the influence of perforation erosion on DAS signal responses. Multiple simulation cases were performed with varying perforation diameters. In Figure 3, the dotted line represents Equation (2), while the solid line represents the calculated results obtained from the CFD simulation. As shown in Figure 3, a linear relationship was observed, consistent with Pakhotina [14,15] and the intercept shifted upward as the perforation diameter increased. From this relationship, the following equation was derived.(2)$$\log\left(q\right)=AL_{sp}+B_{ref}+\frac{7}{4}\log\left(\frac{D}{D_{ref}}\right)$$

Hamanaka et al. [18] applied the acoustic correlation incorporating the perforation erosion effect to a well that underwent multi-cluster stage fracturing, aiming to estimate both the fluid distribution and the post-fracture perforation diameter. In this well, DTS, DAS, and a downhole camera were installed. The estimated fluid distribution was comparable to that obtained from DTS data, and the estimated post-fracture perforation diameters were comparable to the measurements acquired from the downhole camera.

## 3. CFD Acoustic Simulation

### 3.1. Work Flow

The commercial CFD software Ansys Fluent (Ansys 2024) [19] was employed to investigate the acoustic response of perforation flow. A computational geometry was first constructed, where the larger pipe represents the wellbore and a smaller pipe connected at its center represents the perforation as shown in Figure 4a. This geometry was discretized into a computational mesh, which was then used for CFD simulations (Figure 4b).

Prior to the transient calculations, steady-state initialization was performed using the k–ε model. Subsequently, turbulence was calculated using Large Eddy Simulation (LES), while the acoustic source was computed with the Ffowcs Williams-Hawkings model (FW-H model). In actual multi-stage hydraulic fracturing operations, multiple perforations are present, and the interactions between their flows may influence the acoustic signal. In addition, the use of proppant can also affect the acoustic response. According to Chen’s report [20], in addition to the peak generated by the fluid flow, another peak appears at a higher frequency when proppant is used. However, it should be noted that this study is a fundamental investigation focused on the effect of perforation shape on the acoustic signal, and the interactions among multiple perforations as well as the effects of proppant have not been considered.

LES was used to calculate the Reynolds stress and velocity for the fluid flow analysis. Figure 5 presents a schematic diagram of the perforation, illustrating the presence of eddies of various scales. As depicted, eddies larger than the grid size is resolved in LES by discretizing the flow domain and solving the Continuity (Equation (3)) and Navier–Stokes (Equation (4)) equations. In contrast, smaller eddies cannot be directly resolved numerically, and their effects are incorporated through the subgrid-scale term, which is added to the last term of the Navier–Stokes equations.(3)$$\frac{\partial \rho }{\partial t}+\frac{\partial }{\partial x_{i}}\left(\rho \tilde{u_{i}}\right)=0$$(4)$$\frac{d}{dt}\left(\rho \tilde{u_{i}}\right)+\frac{\partial }{\partial x_{j}}\left(\rho \tilde{u_{i}}\tilde{u_{j}}\right)=-\frac{\partial \tilde{p}}{\partial x_{j}}+\frac{\partial \tilde{\sigma_{ij}}}{\partial x_{j}}-\frac{\partial \tau_{ij}}{\partial x_{j}}$$
where ρ is fluid density, u is velocity, p is pressure, $\sigma_{ij}$ is viscous stress tensor, and $\tau_{ij}$ is subgrid-scale stress. 

Following the calculation of Reynolds stress and velocity, the acoustic signal was computed based on these results. Lighthill, a pioneer in aerodynamics, considered the case illustrated in Figure 6, where turbulence, acting as the sound source, is surrounded by an ideal fluid in which the observer is located [21]. 

Equation (5) represents Lighthill’s Acoustic Analogy, which describes the propagation of sound generated by turbulent flow through an ideal fluid. The left-hand side of the equation represents sound propagation in an ideal fluid, while the right-hand side corresponds to the sound source.(5)$$\left(\frac{1}{c_{0}^{2}}\frac{\partial^{2}}{\partial t^{2}}-\nabla^{2}\right)c_{0}^{2}\rho^{\prime }=\frac{\partial^{2}T_{ij}}{\partial x_{i}\partial x_{j}}$$

Lighthill’s stress tensor is defined by the following equation. The first term on the right-hand side represents Reynolds stress, the second term corresponds to wave amplitude nonlinearity and mean density variations, and the third term accounts for attenuation due to viscosity.(6)$$T_{ij}=\rho v_{i}v_{j}+\left(p^{\prime }-c_{0}^{2}\rho^{\prime }\right)\delta_{ij}-\sigma_{ij}$$
where $c_{0}$ is average sound speed, ρ is density, $\rho^{\prime }$ is density fluctuation, $p^{\prime }$ is sound pressure, v is fluid velocity, and σ is viscous stress. 

Ffowcs Williams and Hawkings [22] extended the Lighthill acoustic analogy to consider the effect of surface. Ansys Fluent utilize Equation (7), so the CFD acoustic simulation considers perforation wall effect as well as the turbulence. The left-hand side of the equation represents the wave equation, whereas the right-hand side denotes the acoustic sources expressed in terms of turbulence field variables.(7)$$\begin{array}{cc} \frac{1}{{c_{0}}^{2}}\frac{\partial^{2}p^{\prime }}{{\partial t}^{2}}-\nabla^{2}p^{\prime } & =\frac{\partial }{\partial t}\left[\left[\rho_{0}u_{j}+\rho \left(v_{j}-u_{j}\right)\right]\frac{\partial H}{\partial x_{j}}\right]\\ & -\frac{\partial }{\partial x_{i}}\left[\left[\rho v_{i}\left(v_{j}-u_{j}\right)+\left(p-p_{0}\right)\delta_{ij}-\sigma_{ij}\right]\frac{\partial H}{\partial x_{j}}\right]\\ & +\frac{\partial^{2}}{\partial x_{i}\partial x_{j}}\left[T_{ij}H\left(f\right)\right] \end{array}$$
where $\rho_{0}$ is mean density, $p_{0}$ is mean pressure, $u_{j}$ is the surface velocity, and $H\left(f\right)$ represents the Heaviside function. 

The acoustic pressure obtained from CFD simulations is transformed into a frequency spectrum. The procedure involves three main steps. First, the acoustic pressure is converted into the frequency spectrum using a discrete Fourier transform (DFT). Second, the auto-spectral density is computed, and finally, the sound pressure level and the overall sound pressure level are evaluated from the auto-spectral density. In this study, the overall sound pressure level is used to evaluate the acoustic signals for each case, and the details of each step are explained below.

The acoustic pressure is initially transformed into the frequency-domain signal X through DFT. (8)$$X\left[m\right]=\sum_{n=0}^{N-1}w\left[n\right]p^{\prime }\left[n\right]\exp\left(i2\pi \frac{mn}{N}\right),\;\left(m=1,\;2,\ldots ,\;N\right)$$
where w represents the window function and N denotes the total number of samples. To minimize spectral leakage, a window function is applied, with the Hanning window [23] adopted in this study. The auto-spectral density is then calculated using the following formulation, (9)$$S\left[m\right]=\frac{2X^{*}\left[m\right]X\left[m\right]}{\alpha^{2}},\;\left(m=0,\;1,\;2,\ldots ,\;\frac{N}{2}\right)$$
where the symbol * indicates the complex conjugate and $\alpha =\sum_{n=0}^{N}w[n]$. 

Using S, the sound pressure level is subsequently computed. (10)$$L_{sp}\left[m\right]=10\log\left(\frac{S\left[m\right]}{P_{ref}^{2}}\right),\;\left(m=0,\;1,\;2,\ldots ,\;\frac{f_{s}}{2}\right)$$

The sampling frequency $f_{s}$ determined from the simulation time-step size, governs the maximum resolvable frequency. Fourier transform analysis allows frequencies up to half of the sampling frequency to be reproduced. The overall sound pressure level is obtained by summing the SPL values over a specified frequency band.(11)$$L_{sp_{overall}}=10\log\left(\sum_{m=f_{min}}^{m=f_{max}}10^{\frac{L_{sp}\left[m\right]}{10}}\right)$$
where $f_{min}$ and $f_{max}$ denote the minimum and maximum frequencies of the band, respectively.

### 3.2. Oval-Shaped Perforation Geometry

In contrast to previous studies, the objective of this research is to investigate how irregularly shaped perforations influence the acoustic signal. Therefore, this section describes the perforation geometries used in the simulations. 

As shown in Figure 2, perforations do not erode into a perfectly circular shape but rather take on a form closer to an ellipse. In this study, this behavior is reproduced using the oval geometry available in ANSYS Design Modeler. A schematic of the perforation geometry used is presented in Figure 7a. In Figure 7a, a represents the minor axis and b denotes the major axis. Three perforation shapes were considered in the simulations, defined by the ratio a/b: a/b=1 (Circle), a/b=3/4 (Oval 3/4), and a/b=3/5 (Oval 3/5). These ratios of a/b=3/4 and 3/5 were selected based on the ratios observed in the field and from the image shown in Figure 2 marker ③. To evaluate the effects of both perforation size and shape on the acoustic signal, simulations were conducted for two diameters, a=0.358 inches and a=0.408 inches. For each geometry, cases were simulated at perforation pressure drops of 1000, 1500, and 2000 psi. Figure 7b and Figure 7c present the mesh and the top-view geometry used in the simulations, respectively, confirming the oval shape of the perforations. In general, perforations tend to erode more significantly in the flow direction; therefore, as shown in Figure 7c, the major axis b was oriented parallel to the flow direction.

Perforation erosion proceeds in two stages: the rounding stage, where the perforation entrance becomes rounded, and the stable growth stage, where the perforation size changes [12]. The oval perforations shown in Figure 2 correspond to the stable growth stage, at which point the entrance has already rounded. Therefore, in this study, the perforation entrance was modeled as rounded, as illustrated in Figure 8. Hamanaka [17] reported that when comparing rounded and unrounded perforations, the rounded case exhibited lower acoustic signal levels. This finding suggests that the initial fading of DAS response is attributable to the perforation entrance being rounded by proppant injection.

## 4. Model Validation Using a Circular Perforation

In this section, we conducted simulations using a circular perforation to verify the model employed in this study by examining whether the linear relationship between log(q) and sound pressure level observed in [13,14], and Ref. [15] could be reproduced. Simulations were conducted using a circular perforation. A comparison between the simulation results and the experimental case reported in [13] is presented in Figure 9. From Figure 9, it was confirmed that a linear relationship exists between log(q) and sound pressure level for the circular perforation and the simulation results are in good agreement with the experimental data. Therefore, it was verified that the model functions correctly. However, it should be noted that this study is a fundamental investigation focused on the effect of perforation shape on the acoustic signal, and the interactions among multiple perforations as well as the effects of proppant have not been considered.

## 5. Comparison Under Equal Minor Axis

Here, the simulation results are presented for the case in which the value of the minor axis was fixed while the major axis was varied. Simulations were conducted for two cases. In the first case, the a was set to 0.358 inches, and in the second case, a was set to 0.408 inches. For both cases, the three geometries illustrated in Figure 7a were employed. In the case of a=0.358 inches, the corresponding minor axes b were 0.358, 0.477, and 0.597 inches, whereas for a=0.408 inches, the values of b were 0.408, 0.544, and 0.680 inches. Furthermore, for each geometry, simulations were performed such that the perforation pressure drop was 1000, 1500, and 2000 psi. The results for each case are presented in Figure 10. 

Figure 10a,b present the results for Case 1 and Case 2, respectively, illustrating the relationship between log(q) and the overall sound pressure level. The red arrows indicate how to interpret the graph. The longer arrow represents the relationship for the initial circular perforation before erosion, while the shorter arrow represents the transition after erosion when the perforation becomes oval. In both cases, a linear relationship is observed between log(q) and overall sound pressure level, demonstrating that the sound amplitude increases with increasing flow rate. In addition, as erosion progresses and the major axis of the oval-shaped perforation becomes larger, the intercept increases while maintaining a constant slope. This indicates that, at the same flow rate, a larger perforation size results in a lower overall sound pressure level. For example, in Case 1, when $log\left(q\right)=0.8$, the overall sound pressure level is 107 dB for the Oval 3/4 case, whereas it decreases to 103 dB for the more eroded Oval 3/5 case. These results are consistent with those reported in a previous study that simulated circular perforation erosion.

## 6. Comparison Under Equal Cross-Sectional Area

Here, the influence of oval-shaped perforations is examined through a comparison with circular perforations of equal cross-sectional area. In Case 3, a comparison is made between a circular perforation with a diameter of 0.358 inches and an Oval 3/4 perforation with a=0.3 inches and, b=0.4 inches. In this case, both perforations have an identical cross-sectional area of $0.101\;{in}^{2}$. In Case 4, a circular perforation with a diameter of 0.408 inches is compared with an Oval 3/5 perforation with a=0.3 inches and, b=0.5 inches, where both perforations have a cross-sectional area of $0.131{\;in}^{2}$. For each geometry, simulations were performed such that the perforation pressure drop was 1000, 1500, and 2000 psi. The results for each case are presented in Figure 11. 

Figure 11a,b show the results of Case 3 and Case 4, respectively, each depicting the relationship between log(q) and the overall sound pressure level. In both cases, it is observed that the intercept is larger for the oval-shaped perforations compared to the circular perforations with the same cross-sectional area. This indicates that, at an equal flow rate, the overall sound pressure level is higher for oval-shaped perforations than for circular ones. For example, in Case 3, when $log\left(q\right)=0.5$, the overall sound pressure level is 101.6 dB for the circular perforation, whereas it increases to 104.1 dB for the oval-shaped perforation. The acoustic responses at $log\left(q\right)=0.5$, and flow rates at overall sound pressure level =100 dB for each perforation shape in Case 3 and Case 4 are summarized in Table 1. When the overall sound pressure level is 100 dB, the flow rate error is approximately 17% in Case 3 and 14% in Case 4. This indicates that even when the same amplitude DAS signal is obtained, assuming circular perforations when estimating the fluid distribution leads to an overestimation of about 14–17% if the actual perforation shape is oval. Since this deviation is not negligible, a correction is required, and this issue is further discussed in the section on Acoustic Correlation. Even when the orifice cross-sectional area is the same, the ease of fluid flow varies depending on its geometry. As reported in [24], noncircular shapes (e.g., rectangular) tend to cause more flow disturbance and greater energy dissipation compared with circular ones. This occurs because the presence of corners and abrupt geometric transitions promotes flow separation. The oval-shaped perforation used in this study, shown in Figure 7a, consists of semicircular ends and a straight middle section. Although the geometric transition is less abrupt than that of a rectangular shape, there are still regions at the junctions between the semicircular and straight segments where the curvature changes discontinuously. These local geometric variations disturb the flow direction, inducing vortices and recirculation zones, which increase energy dissipation. Consequently, the oval perforation restricts the fluid flow more than the circular one, leading to the generation of a stronger sound pressure. Based on these results, it is suggested that assuming circular perforations when estimating fluid distribution from DAS signal could lead to overestimations or inaccuracies, especially when the actual perforation shape is oval. Such inaccuracies could result in misinterpretation of fluid allocation among clusters or stages, potentially leading to incorrect assessment of stimulation efficiency in field-scale DAS analysis.

## 7. Acoustic Correlation

Here, we investigate whether Equation (2) derived by Hamanaka [17] can also be applied to irregularly shaped perforations. For this investigation, two additional geometries, Oval 3/6 and Oval 3/7, as shown in Figure 12, were prepared. In Oval 3/6, the ratio of a/b is 3/6, while in Oval 3/7, the ratio of a/b is 3/7. For both cases, a value of a=0.358 inches was used. The corresponding major axis lengths b are 0.716 inches and 0.835 inches, respectively.

Equation (2) evaluates the impact of erosion on the acoustic signal using the diameter of an eroded perforation. However, since this study employs oval-shaped rather than circular perforations, the equivalent diameter was calculated and used for the analysis. The equivalent diameter was determined using $D_{eq}=\sqrt{\frac{4\times Area}{\pi }}$. The equivalent diameters $D_{eq}$ of the oval perforations were calculated as follows: Oval 3/4, 0.427 inches; Oval 3/5, 0.487 inches; Oval 3/6, 0.540 inches; and Oval 3/7, 0.588 inches. 

The results are shown in Figure 13. The solid lines represent the plotted simulation results, while the dashed lines indicate the correlation. The correlation was calculated by adding $\frac{7}{4}log\left(D_{eq}/D_{ref}\right)$ corresponding to each a/b ratio to the intercept of the circular line. As shown in Figure 13, the correlation closely matches the simulation results for all a/b ratios, indicating that Equation (2) is applicable to oval-shaped perforations as well. However, as illustrated in Figure 11, it should be noted that when the area is the same, oval-shaped perforations result in a higher overall sound pressure level compared to circular perforations.

However, since Equation (2) was derived for circular perforations, it does not account for variations in perforation shape. Therefore, it is rewritten in a form based on the cross-sectional area. $$\log\left(q\right)=AL_{sp}+B_{ref}+\frac{7}{8}\log\left(\frac{A_{perf}}{A_{ref}}\right)$$(12)$$\Leftrightarrow \log\left(q\right)=AL_{sp}+B_{ref}+\frac{7}{8}\log\left\{\frac{\left(\frac{\pi a^{2}}{4}+a\left(b-a\right)\right)}{\left(\frac{\pi a^{2}}{4}\right)}\right\}$$
where $A_{perf}$ denotes the area of the eroded perforation, and $A_{ref}$ is the area of the circular perforation.

When expressed with area, the original diameter-based term changes so that the denominator becomes 8, because area scales with the square of the diameter. Here, we examine whether Equation (12) is applicable to oval-shaped perforations. The results are shown in Figure 14. The solid lines represent the plotted simulation results, while the dashed lines indicate the correlation. The correlation was calculated by adding $\frac{7}{8}log\left(\left(\frac{\pi a^{2}}{4}+a\left(b-a\right)\right)/\left(\frac{\pi a^{2}}{4}\right)\right)$ corresponding to each a/b ratio to the intercept of the circular line. As shown in Figure 14, the correlation closely matches the simulation results for all a/b ratios, indicating that Equation (12) is applicable to oval-shaped perforations. In the case of Equation (2), the use of diameter is required, which makes it impossible to account for the effect of perforation shape. However, Equation (12) incorporates both the minor and major axes of an oval-shaped perforation, allowing the influence of perforation geometry to be considered. Therefore, by applying Equation (12), the potential errors in DAS interpretation caused by differences in perforation shape, as discussed in Chapter 6, can be mitigated.

## 8. Conclusions

This study established a theoretical framework for simulating the overall sound pressure level using CFD, enabling a deeper understanding of the acoustic response generated during hydraulic fracturing treatments. The analysis particularly emphasized the impact of perforation erosion, focusing on oval-shaped geometries in comparison with circular perforations. The main conclusions are as follows:

From the results of Case 1 and Case 2, it was confirmed that even when perforations erode in the flow direction and evolve into oval shapes, the relationship between log(q) and the overall sound pressure level continues to follow a straight line with a defined intercept and slope, consistent with previous studies.From the results of Case 1 and Case 2, it was also shown that even when perforations erode into an oval shape and only the major axis increases, the intercept increases logarithmically with erosion, resulting in a reduction in sound amplitude, while the slope remains constant.From the results of Case 3 and Case 4, it was demonstrated that when the cross-sectional area and flow rate are the same, oval-shaped perforations generate a greater overall sound pressure level compared to circular perforations. Based on these results, it is suggested that when estimating flow rate from DAS data, assuming a circular erosion shape could lead to overestimations or inaccuracies.By rewriting the correlation derived by Hamanaka [17] in terms of the cross-sectional area, an area-based correlation was developed that incorporates both the minor and major axes of oval-shaped perforations. This area-based correlation enables consideration of the effects of perforation shape differences and helps reduce potential errors in DAS interpretation that may arise from variations in perforation geometry.

## Figures and Tables

**Figure 1 sensors-25-07037-f001:**
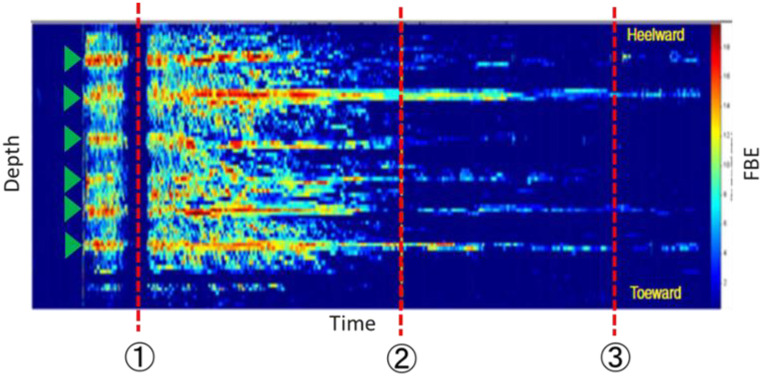
Decay of DAS signal due to perforation erosion (modified from [11]).

**Figure 2 sensors-25-07037-f002:**
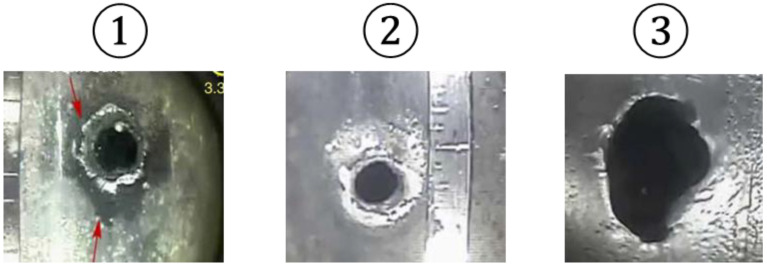
Perforation Shape Changes with Erosion (after [12]).

**Figure 3 sensors-25-07037-f003:**
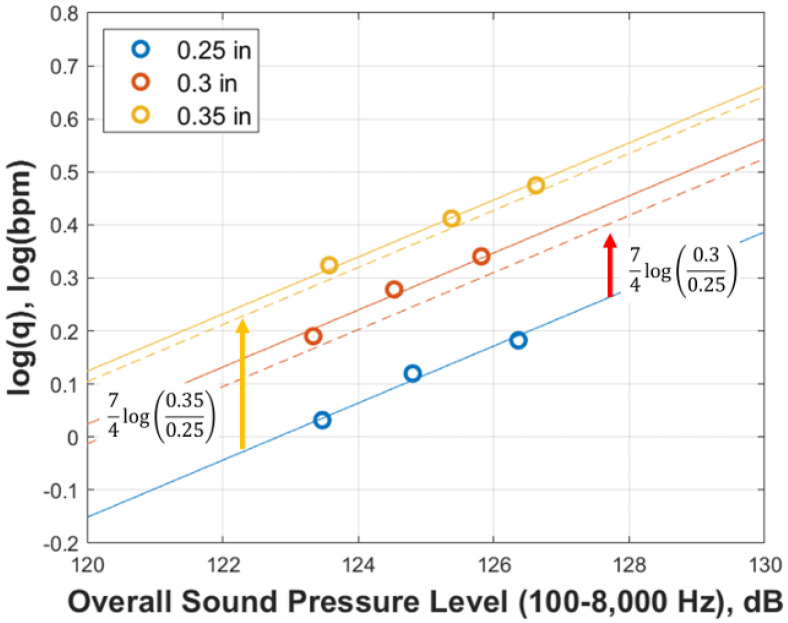
Acoustic correlation compensating perforation erosion effect (reprinted from [17]).

**Figure 4 sensors-25-07037-f004:**
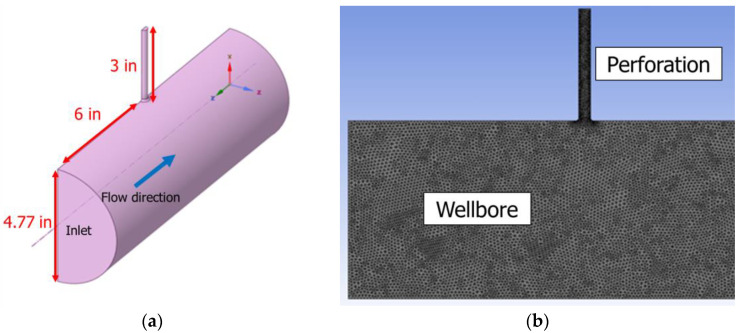
CFD Acoustic simulation domain: (**a**) geometry, and (**b**) mesh.

**Figure 5 sensors-25-07037-f005:**
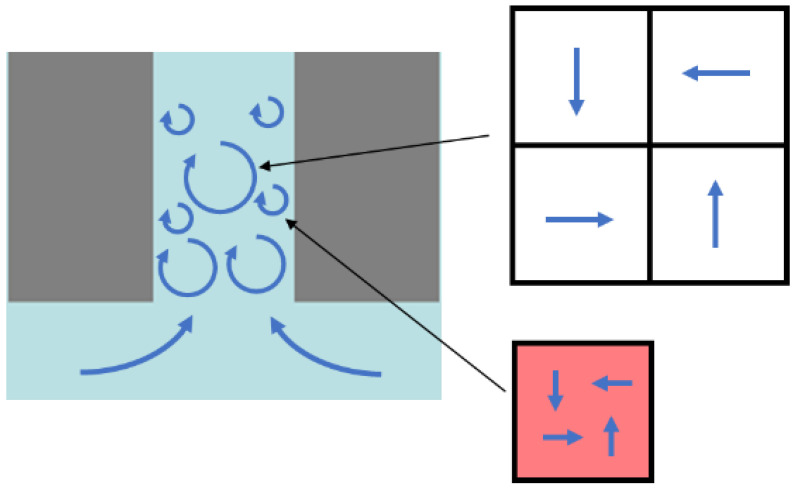
Schematic diagram of the perforation for LES.

**Figure 6 sensors-25-07037-f006:**
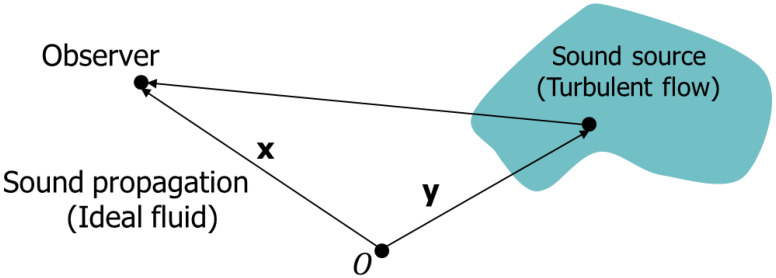
Lighthill’s acoustic analogy configuration (reprinted from [17]).

**Figure 7 sensors-25-07037-f007:**
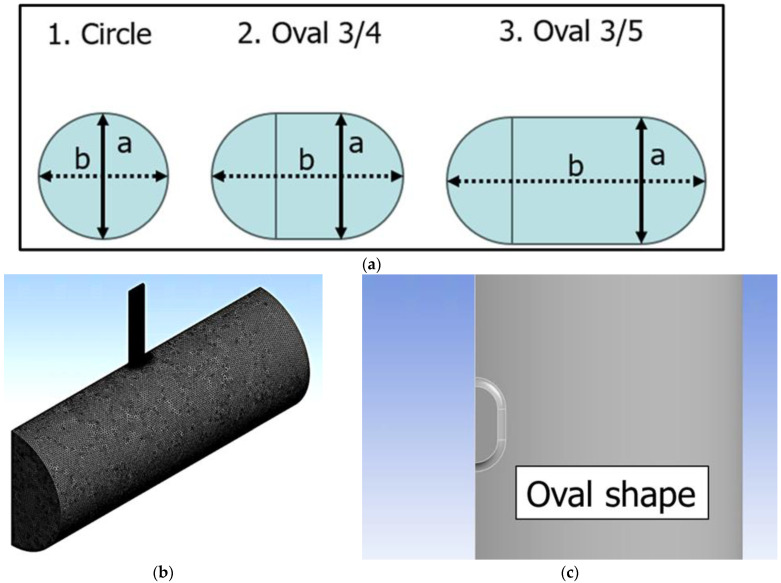
Simulated perforation shapes: (**a**) perforation shapes, (**b**) used mesh, (**c**) used geometry (top view).

**Figure 8 sensors-25-07037-f008:**
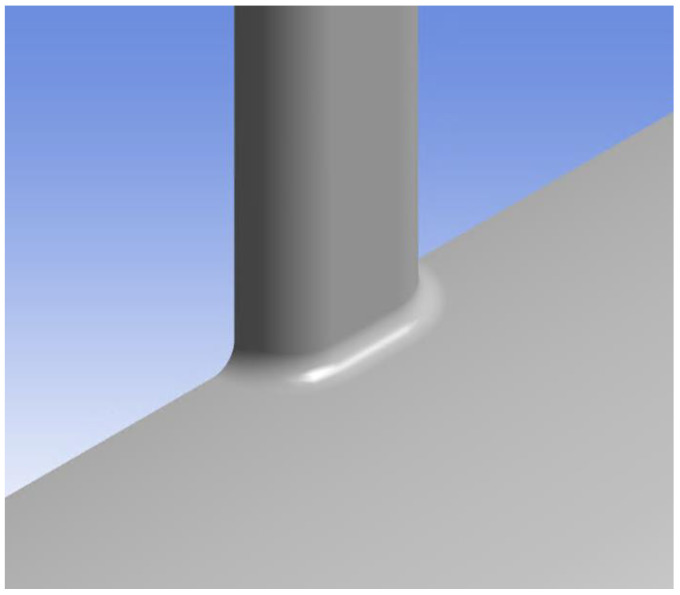
CFD geometry at perforation entrance.

**Figure 9 sensors-25-07037-f009:**
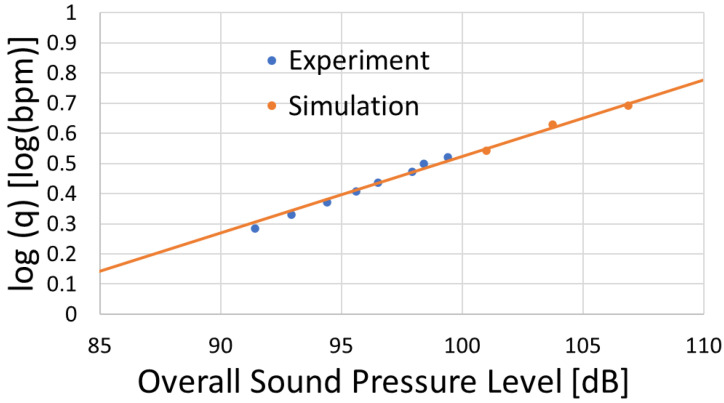
Acoustic correlation between log(q) and sound pressure level for circular perforation.

**Figure 10 sensors-25-07037-f010:**
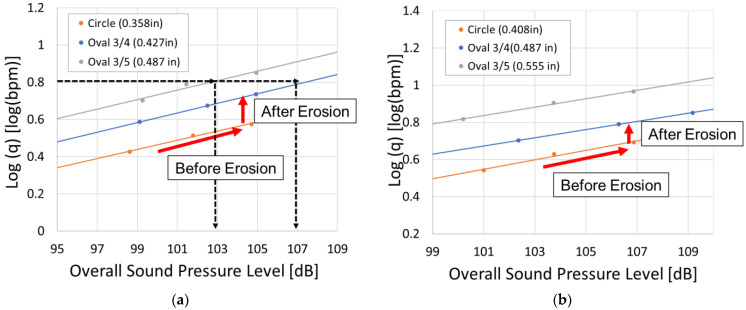
Simulation results for comparison under equal minor axis: (**a**) case 1: minor axis a=0.358 inches, (**b**) case 2: a=0.408 inches.

**Figure 11 sensors-25-07037-f011:**
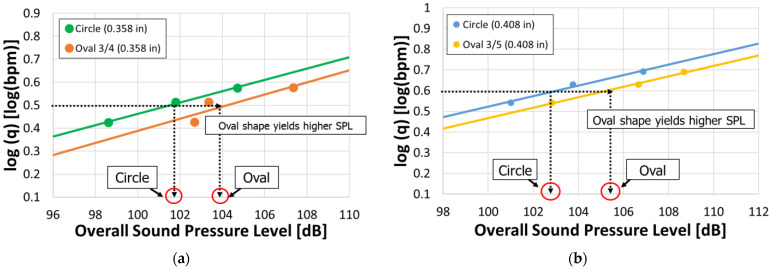
Simulation results for comparison under cross-sectional area: (**a**) case 3: cross-sectional area is $0.101\;{in}^{2}$, (**b**) case 4: cross-sectional area is $0.131\;{in}^{2}$.

**Figure 12 sensors-25-07037-f012:**
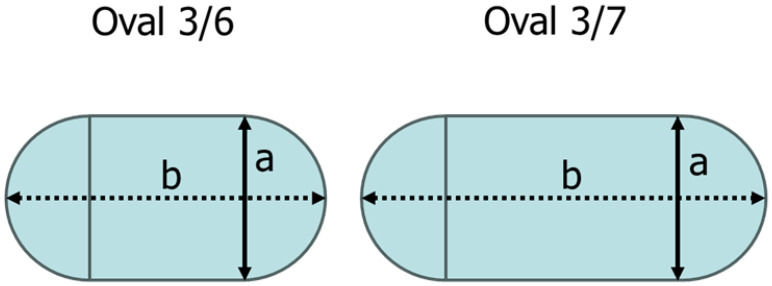
Additionally simulated perforation shapes.

**Figure 13 sensors-25-07037-f013:**
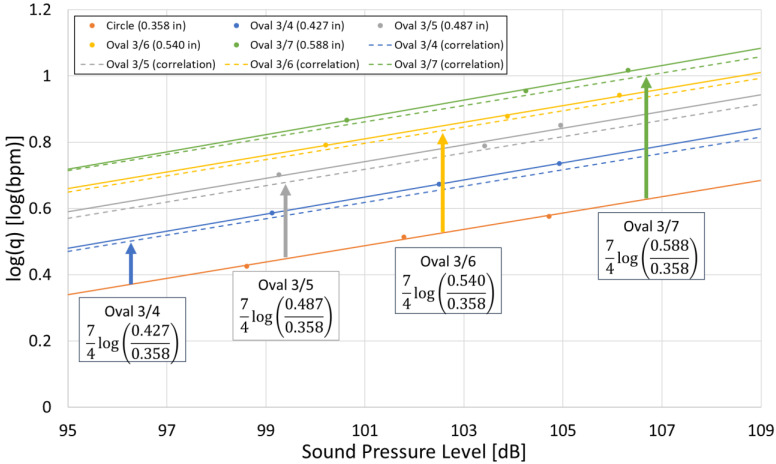
Diameter-based correlation and simulation results for oval perforations with various a/b ratios. The solid line represents the simulation results, while the dashed line indicates the correlation. The results are shown for Circle, Oval 3/4, Oval 3/5, Oval 3/6, and Oval 3/7.

**Figure 14 sensors-25-07037-f014:**
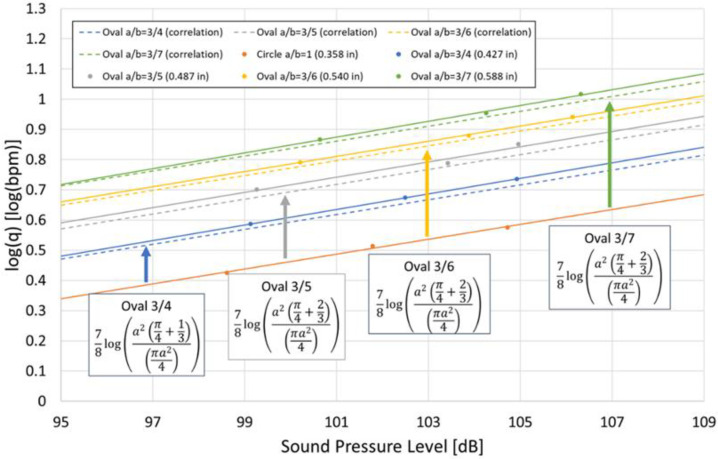
Area-based correlation and simulation results for oval perforations with various a/b ratios. The solid line represents the simulation results, while the dashed line indicates the correlation. The results are shown for Circle, Oval 3/4, Oval 3/5, Oval 3/6, and Oval 3/7.

**Table 1 sensors-25-07037-t001:** Acoustic response of circular vs. oval perforations under equal cross-sectional area.

Cases	Perforation Shapes	Overall Sound Pressure Level (dB) at Log(q) = 0.5	Flow Rate (BPM) at Overall Sound Pressure Level = 100 (dB)
Case 3 *^1^	Circle	101.6	2.88
	Oval 3/4	104.1	2.46
Case 4 *^2^	Circle	99.0	3.35
	Oval 3/5	101.2	2.95

*^1^ Cross-sectional area = $0.101\;{in}^{2}$, *^2^ Cross-sectional area = $0.131\;{in}^{2}$.

## Data Availability

The original contributions presented in the study are included in the article, further inquiries can be directed to the corresponding author.

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
