# Peer review of "Evaluating Fluid Distribution by Distributed Acoustic Sensing (DAS) with Perforation Erosion Effect"

_sensors, 2025, doi:10.3390/s25227037_

Round 1

Reviewer 1 Report

Comments and Suggestions for Authors

The manuscript presents a well-structured and technically sound investigation into the influence of perforation erosion on Distributed Acoustic Sensing (DAS) signals during hydraulic fracturing. The use of CFD simulations with oval-shaped perforations to mimic real-world erosion patterns is a noteworthy contribution. However, the manuscript would benefit from clearer articulation of its novelty and a more in-depth discussion of the practical implications of the findings.

Comments

  1. Please further highlight the novelty in the Introduction and Abstract. Explicitly state that using oval perforations, rather than the circular ones common in prior studies, provides more realistic modeling. Clarify how this offers new insights for interpreting field DAS data.
  2. Please provide justification for the selected oval ratios (a/b=3/4, 3/5, etc.). A brief explanation or reference to field data supporting these specific geometry choices would strengthen the methodological foundation of the simulation study.
  3. For clarity, please enhance Figures 9, 10, and 12 captions by explicitly defining "Case 1" and "Case 2". Consider adding a summary table to directly compare the acoustic response of circular vs. oval perforations under equal cross-sectional area.
  4. The discussion should be expanded to explain the physical mechanism causing higher sound levels in oval perforations at equal area. Also, contextualize the practical implications of potential inaccuracies from assuming circular erosion in current DAS analysis.
  5. Minor Issues

Abstract: The phrase “only the intercept shifts upward logarithmically” could be rephrased for clarity.

Keywords: Consider adding “CFD simulation” or “perforation geometry” to improve searchability.

References: Ensure all citations are consistently formatted (e.g., [8] Ansys Fluent theory guide).

Proofreading: Minor grammatical issues exist (e.g., “This strategy aligns with the limited-entry completion concept, in which fewer perforations are used…”).

Overall, this is a valuable study that advances the understanding of DAS signal interpretation in eroded perforations. With minor revisions, it will be suitable for publication.

Author Response

Comments 1: [Please further highlight the novelty in the Introduction and Abstract. Explicitly state that using oval perforations, rather than the circular ones common in prior studies, provides more realistic modeling. Clarify how this offers new insights for interpreting field DAS data.]

Response 1: Thank you for pointing this out. We agree with this comment. Therefore, we have [added the text emphasizing the use of oval-shaped perforations instead of the conventional circular ones. First, in the Abstract section, we added the sentence "which provides more realistic modeling." at line 21, and “This indicates that even when the same amplitude DAS signal is obtained, assuming circular perforations when estimating the fluid distribution leads to an overestimation if the actual perforation shape is oval.” at lines 32–34 on page 1.
Then, in the Introduction section, we used the sentence “Although previous studies have investigated the influence of perforation erosion on DAS signals, those studies assumed circular perforation geometries, and the effect of irregular shapes on DAS signal response is poorly understood. Therefore, in this study, we reproduce the perforation geometry observed in the field using oval-shaped perforations and investigate its effect for more realistic DAS interpretation.” at lines 101–105 on page 3 to emphasize the difference from previous studies.

Comments 2: [Please provide justification for the selected oval ratios (a/b=3/4, 3/5, etc.). A brief explanation or reference to field data supporting these specific geometry choices would strengthen the methodological foundation of the simulation study.] 

Response 2: Thank you for pointing this out. We agree with this comment. Therefore, we have [added the explanation at Page 7, Lines 264-265]

Comments 3: [For clarity, please enhance Figures 9, 10, and 12 captions by explicitly defining "Case 1" and "Case 2". Consider adding a summary table to directly compare the acoustic response of circular vs. oval perforations under equal cross-sectional area.]

Response 3: Thank you for pointing this out. We agree with this comment. Therefore, we have [ modified captions at Page 10, Lines 317-318; Page 11, Lines 342-343; and Page 13 Lines 407-409, respectively. We have added summary table at Page 12, Line 377]

Comments 4: [The discussion should be expanded to explain the physical mechanism causing higher sound levels in oval perforations at equal area. Also, contextualize the practical implications of potential inaccuracies from assuming circular erosion in current DAS analysis.]

Response 4: Thank you for pointing this out. We agree with this comment. Therefore, we have expanded the discussion section (Page 11-12, Lines 359-375) to explain the physical mechanism responsible for the higher sound pressure levels observed in oval perforations compared to circular ones at equal cross-sectional area.

Comments 4: [Minor Issues

Abstract: The phrase “only the intercept shifts upward logarithmically” could be rephrased for clarity.

Keywords: Consider adding “CFD simulation” or “perforation geometry” to improve searchability.

References: Ensure all citations are consistently formatted (e.g., [8] Ansys Fluent theory guide).

Proofreading: Minor grammatical issues exist (e.g., “This strategy aligns with the limited-entry completion concept, in which fewer perforations are used…”).

Overall, this is a valuable study that advances the understanding of DAS signal interpretation in eroded perforations. With minor revisions, it will be suitable for publication.]

Response 4: Thank you for pointing this out. We agree with this comment. Therefore, we have [ modified them at Page 1, Line 28; Page 1, Line 38; Page 15-16, Lines 480-539; and Page 2, Lines 57-59]

Reviewer 2 Report

Comments and Suggestions for Authors

The article, entitled «Evaluating Fluid Distribution by Distributed Acoustic Sensing (DAS) with Perforation Erosion Effect», is devoted to the topical problem of interpreting distributed acoustic logging data during hydraulic fracturing, taking into account the real, non-ideal geometry of perforation holes subject to erosion. The primary contribution of the work lies in the implementation of a comprehensive numerical modelling approach, utilizing oval perforation shapes. This approach offers a more precise reflection of field observations in comparison to the commonly accepted assumption of perfectly circular erosion. It has been demonstrated that perforation geometry exerts a significant effect on the acoustic signal. Furthermore, the correlations identified can be extended to the case of oval erosion.

This study addresses the fundamental problem of optimizing multistage hydraulic fracturing by focusing on the control of fluid distribution between perforation clusters. The utilization of fiber optic methodologies DAS to oversee this process constitutes a state-of-the-art approach. A comprehensive understanding of the impact of perforation erosion on the DAS signal is imperative for accurate data interpretation. The primary benefit of this study is its deviation from the conventional assumption of round perforations. The modeling of oval geometries, which are more suitable for real conditions, represents a significant advancement compared to previous studies.

The work is logically structured, beginning with an examination of the effect of erosion at a fixed minor axis and concluding with a comparison of perforations with the same cross-sectional area. This methodological approach enables the consistent identification of the impact of hole size and shape on the acoustic signal. The work provides a high-quality visual representation: The geometries, grids, and simulation results are presented in clear figures to facilitate understanding of the analysis.

The following comments and recommendations are provided for consideration:

1) The paper does not verify the results of CFD modeling against controlled physical experiments or field data. The authors cite the works of Chen K. [4] Pakhotina I. [5], and Pakhotina J. et al. [6] which establish a linear correlation. However, they do not directly compare their own modeling results with their experimental data to verify their specific model. The authors should include a section devoted to the validation of their numerical model. It is imperative to demonstrate that the model replicates established empirical dependencies, as evidenced in prior studies such as [4, 5, 6], for the fundamental case of circular perforation prior to advancing to the analysis of oval geometries. Absent this fundamental element, the credibility of the entire set of results is called into question.

2) However, a critical examination of the article reveals a conspicuous omission: a discussion of the primary limitations of the approach employed. The present simulation is performed for a single perforation; however, in a real well, there is interaction between flows from multiple clusters. Furthermore, the model does not consider the presence of proppant in the fluid, which is the primary cause of erosion and can substantially impact the acoustic characteristics of the flow.

3) The conclusion that the assumption of circular erosion can lead to underestimations is important, but it is not developed. It is imperative to ascertain the precise manner in which an engineer should adjust the interpretation of DAS data. The magnitude of this error must be assessed quantitatively to determine its significance. The authors are tasked with the development of this conclusion. It would be advantageous to provide practical guidance or a correction factor that would enable the consideration of perforation ovality when calculating flow velocity based on DAS signal amplitude.

4) To apply the Hamanaka correlation [7] to oval perforations, the authors utilize the hydraulic diameter (Deq =√(4*Area/π)). However, for acoustic phenomena that are highly dependent on the geometry of the source and boundary layer, this may not be entirely accurate. The selection of this specific definition of diameter is not supported by compelling evidence. The authors should provide a concise justification for the selection of the hydraulic diameter as equivalent. A subsequent discussion would be beneficial in determining whether this parameter is the most suitable for describing acoustic emission, or if other parameters (e.g., perimeter) could offer a superior fit.

5) As illustrated in Figure 4, the direction of fluid flow within the casing aligns with the larger radius of the oval-shaped opening (b). The data presented in the paper does not provide sufficient information to draw conclusions about how the simulation results will change when the orientation of the oval perforation opening relative to the direction of fluid flow is altered.

6) In order to enhance the relevance of the work in the field, it is recommended that an overview of the state of the art be added, supported by recent publications. In the present version, 11 cited publications appear to be inadequate, with the majority having been published in the distant past.

Notwithstanding, the study presented here possesses considerable scientific potential and addresses a significant applied problem. The innovative approach with oval erosion modeling warrants commendation. However, in its current form, the article suffers from a lack of validation of the numerical model and insufficiently in-depth analysis of its limitations and practical implications. The article would benefit from additional refinement.

Author Response

Comments 1: [The paper does not verify the results of CFD modeling against controlled physical experiments or field data. The authors cite the works of Chen K. [4] Pakhotina I. [5], and Pakhotina J. et al. [6] which establish a linear correlation. However, they do not directly compare their own modeling results with their experimental data to verify their specific model. The authors should include a section devoted to the validation of their numerical model. It is imperative to demonstrate that the model replicates established empirical dependencies, as evidenced in prior studies such as [4, 5, 6], for the fundamental case of circular perforation prior to advancing to the analysis of oval geometries. Absent this fundamental element, the credibility of the entire set of results is called into question.]

Response 1: Validation is important. We have set our model domain similar to the experiment conducted before to validate the model results. We have added a validation section on Page 9, Line 289, comparing the simulation results obtained using circular perforations with the experimental results conducted by Chen.

Comments 2: [ However, a critical examination of the article reveals a conspicuous omission: a discussion of the primary limitations of the approach employed. The present simulation is performed for a single perforation; however, in a real well, there is interaction between flows from multiple clusters. Furthermore, the model does not consider the presence of proppant in the fluid, which is the primary cause of erosion and can substantially impact the acoustic characteristics of the flow.]

Response 2: To address the comment, we have [added a statement regarding the limitations of this study in Section 3.1 (Workflow) on Page 5, Lines 166–169: “However, it should be noted that this study is a fundamental investigation focused on the effect of perforation shape on the acoustic signal, and the interactions among multiple perforations as well as the effects of proppant have not been considered.”

Comments 3: [The conclusion that the assumption of circular erosion can lead to underestimations is important, but it is not developed. It is imperative to ascertain the precise manner in which an engineer should adjust the interpretation of DAS data. The magnitude of this error must be assessed quantitatively to determine its significance. The authors are tasked with the development of this conclusion. It would be advantageous to provide practical guidance or a correction factor that would enable the consideration of perforation ovality when calculating flow velocity based on DAS signal amplitude.]

Response 3: We have added a discussion on Page 11, Lines 352–359, comparing the flow rates of oval and circular perforations with equal cross-sectional areas at the same sound pressure level, and evaluating the resulting errors. The calculation showed that the error ranged from 14% to 17%. Accordingly, we added a new discussion in Section 7 (Acoustic Correlation) on Pages 13–14, Lines 411–438, addressing the development of a new correlation that accounts for the effect of perforation geometry.

Comments 4: To apply the Hamanaka correlation [7] to oval perforations, the authors utilize the hydraulic diameter (Deq =√(4*Area/π)). However, for acoustic phenomena that are highly dependent on the geometry of the source and boundary layer, this may not be entirely accurate. The selection of this specific definition of diameter is not supported by compelling evidence. The authors should provide a concise justification for the selection of the hydraulic diameter as equivalent. A subsequent discussion would be beneficial in determining whether this parameter is the most suitable for describing acoustic emission, or if other parameters (e.g., perimeter) could offer a superior fit.

Response 4: the comment is address with a new discussion in Section 7 (Acoustic Correlation) on Pages 13–14, Lines 411–438, addressing the development of a new correlation that accounts for the effect of perforation geometry.

Comments 5: As illustrated in Figure 4, the direction of fluid flow within the casing aligns with the larger radius of the oval-shaped opening (b). The data presented in the paper does not provide sufficient information to draw conclusions about how the simulation results will change when the orientation of the oval perforation opening relative to the direction of fluid flow is altered.

Response 5: Erosion happens as a result of dynamic turbulence, which could be in any direction. But in the problem that is modeled, the flow is confined in a 6-in diameter long pipe with fluid velocity extremely high, it is reasonable to assume that perforations tend to erode more significantly in the flow direction. We have looked at some downhole impages, which support the assumption. In some special case, perforation could erode perpendicular to the flow direction, this is possible when the initial perforation is not perfectly rounded, with a long er axis in the direction that is perpendicular to flow. For this study, since the goal is to understand the erosion phenomenon therefore, we used a more ideal case. In the future we may investigate the phenomenon if the erosion does not agline with the flow direction. in this study, we only discuss the case in which erosion occurs more strongly in the flow direction as shown in Figure 7(c). We added a sentence explaining this in Section 3.2 on Page 8, Lines 270–272.]

Comments 6: [In order to enhance the relevance of the work in the field, it is recommended that an overview of the state of the art be added, supported by recent publications. In the present version, 11 cited publications appear to be inadequate, with the majority having been published in the distant past.]

Response 6: Thank you for pointing this out. We agree with this comment. Therefore, we have [added explanations about recent studies in the Introduction section on Page 2, Lines 66–74 and Lines 78–86. Lines 66–74 describe the field-observed trends in perforation erosion, while Lines 78–86 explain practical field applications of DAS technology.]